# Defect Structure of Nanocrystalline NiO Oxide Stabilized by SiO₂

Maxim D. Mikhnenko, Svetlana V. Cherepanova, Evgeny Yu Gerasimov, Alena A. Pochtar,
Maria V. Alekseeva (Bykova), Roman G. Kukushkin, Vadim A. Yakovlev and Olga A. Bulavchenko *

Boreskov Institute of Catalysis, Siberian Branch of the Russian Academy of Science, Lavrentiev Ave. 5,
Novosibirsk 630090, Russia
* Correspondence: obulavchenko@catalysis.ru

**Abstract:** In this paper, structural features of the NiO-SiO₂ nanocrystalline catalyst synthesized by the sol-gel method were studied by X-ray diffraction (XRD), high-resolution transmission electron microscopy (TEM), and differential dissolution (DD). The XRD pattern of NiO-SiO₂ significantly differs from the "ideal" NiO pattern: the peaks of the NiO-like phase are asymmetric, especially the 111 diffraction peak. The NiO-SiO₂ nanocrystalline catalyst was investigated by means of XRD simulations based on two approaches: conventional Rietveld analysis and statistical models of 1D disordered crystals. Through a direct simulation of XRD profiles, structural information is extracted from both the Bragg and diffuse scattering. XRD simulations showed that the asymmetry of all the diffraction peaks is due to the presence of two NiO-like oxides with different lattice constants and different average sizes: ~90 wt% of mixed Ni-Si oxide (Ni:Si = 0.14:0.86) with average crystallite sizes (D ~ 27.5 Å) and ~10 wt% of pure NiO (D ~ 50 Å). The high asymmetry of the 111 diffraction peak is due to the appearance of diffuse scattering caused by the inclusion of tetrahedral SiO₂ layers between octahedral NiO layers. Such methods as TEM and DD were applied as independent criteria to prove the structural model, and the results obtained confirm the formation of mixed Ni-Si oxide.

**Keywords:** defect structure; modeling; nanoparticles; nickel oxide

## 1. Introduction

The great interest in nickel-based systems is mainly related to the availability of nickel in the Earth's crust, its high activity in hydrogenation reactions, and its low cost. Nickel-based catalysts are widely used in different industrial processes such as petroleum refining, stream reforming, the synthesis of fine chemicals, etc. [1–7]. In particular, they are studied in the processing of various types of organic materials, such as bio-oil [8] and microalgal lipids, as well as model compounds [2,6,9].

Ni-based catalysts can be classified into several types: (1) supported Ni-based catalysts on alumina, titanium, etc. [2–6,10–14]; (2) mixed oxides such as perovskites [15], aluminates [16], and Ce-Zr oxide based on the fluorite structure [17]; and (3) high Ni-loaded catalysts [18–20]. The latter ones have been extensively studied as catalysts for bio-oil upgrading [9,10,21,22] and methane decomposition [20]. To obtain a high nickel content, a special synthesis approach was developed. In order to stabilize the dispersed active component, Ermakova et al. [23] prepared high-Ni-loaded catalysts by the heterophase sol–gel method using such textural promoters as SiO₂, Al₂O₃, MgO, TiO₂, and ZrO₂. The highest level of metal particle protection against sintering during the reduction is achieved in the presence of 10 wt% SiO₂. The NiO-SiO₂ catalysts prepared by the heterophase sol–gel method had shown their advantages over the impregnated Ni-based catalysts supported on Al₂O₃ or CeO₂-ZrO₂ in the hydrodeoxygenation of guaiacol [24,25].

Improved properties of the NiO-SiO₂ catalyst are determined by the structure and microstructure characteristics of nickel oxide. The XRD pattern of the high-Ni-loaded

NiO-SiO$_2$ catalyst differs from the pattern for bulk nickel oxide. Typically, XRD pattern of the NiO-SiO$_2$ catalyst exhibits broad peaks and their asymmetric shape [24,25]. A rough estimation of the size of NiO nanoparticles based on the Scherrer equation gives a value of 2–3 nm [24]. It is worth emphasizing that the observed diffraction effect cannot be explained only by the very small size of nickel oxide particles and, apparently, is directly related to the presence of Si as the modifying additive. The promoter, on the one hand, stabilizes the small size of NiO particles and, on the other hand, does not prevent the formation of nickel metal nanoparticles during the catalyst activation with hydrogen [26]. In the case of the highly dispersed NiO-SiO$_2$ catalyst, conventional X-ray diffraction methods based on the analysis of peak positions and intensities are inapplicable. Total scattering X-ray methods seem to be effective instruments to study the structure of nanosized nickel oxide particles in the catalysts. The development of computational facilities in the last few years has made it possible to perform a direct calculation of total X-ray scattering from models of nanocrystalline particles. The recently developed method for calculating X-ray diffraction patterns using the Debye formula [27] is well suited for defect-free nanocrystals of different shapes, while in the presence of planar defects, averaging over a large ensemble with their different locations in each particle is necessary, which is laborious and requires significant computational costs. Another approach is a probabilistic method for the analysis of 1D disordered nanocrystals, where the calculation of the XRD pattern is based on a statistical model for the arrangement of planar defects [27]. In particular, inclusions of layers with a structure different from that of the matrix (host) can be considered as planar defects [28–31]. In this case, the variable parameter (apart from the structure of the guest layer) is the fraction of guest layers. The method also makes it possible to take into account the anisotropic shape and size of nanoparticles.

The purpose of this work is to determine the structural features of high-Ni-loaded NiO-SiO$_2$ catalysts prepared by the sol-gel method. The simulation of XRD patterns based on the statistical models of nanocrystalline Ni-Si oxide particles was applied. The high-resolution TEM with energy dispersive X-ray analysis (EDX) and the DD technique were used to determine the distribution of Si on the catalyst. The elucidation of the NiO-SiO$_2$ structure will clarify the mechanism of modification of nickel oxide with the addition of silicon and the stabilization of small particle sizes.

## 2. Results and Discussion

### 2.1. Rietveld Analysis

The XRD pattern of the sample shows broad peaks located at 2θ = 36.5, 43, 62, 75, and 79°, which correspond to cubic or rhombohedral NiO (PDF #47-1049 and PDF #44-1159, respectively) (Figure 1). However, it should be noted that the 111, 220, and 220 diffraction peaks are asymmetric. The greatest asymmetry is observed for the 111 peak and looks like a shoulder on the small-angle side of the peak. Peaks of cubic Ni$_2$SiO$_4$ (PDF #15-0255) are not observed on the experimental diffraction pattern.

Rietveld analysis was carried out on the basis of four models: (a) cubic NiO (Fm3m); (b) rhombohedral NiO (R-3m); (c) two cubic NiO with different lattice constants; (d) cubic NiO and Ni$_2$SiO$_4$ (Fd3m) (Figure 2). The results are shown in Table 1. For cubic NiO, the refined parameter a = 4.227 Å is higher than that for NiO (PDF#47-1049, a = 4.177 Å). In the case of the cubic NiO model, the average crystallite size is ~20 Å. One can see that the intensity of the experimental 111 peak is higher compared to the calculated one (Figure 2a). According to the R-factor, the XRD pattern calculated for rhombohedral NiO modification better corresponds to the experiment relative to cubic NiO, but the calculated 111 peak has a lower intensity than the experimental one (Figure 2b). The XRD pattern simulated for two types of NiO particles having different cubic lattice constants a = 4.169 Å and a = 4.289 Å (Figure 2c) looks like the XRD pattern simulated for the one cubic NiO phase (a = 4.227 Å). Two phases, cubic NiO and cubic Ni$_2$SiO$_4$, give better correspondence with the experimental 111 diffraction peak, but refined lattice constant a = 8.280Å of Ni$_2$SiO$_4$ is noticeably higher relative to the literature data for Ni$_2$SiO$_4$ (PDF#15-0255, a = 8.044 Å).

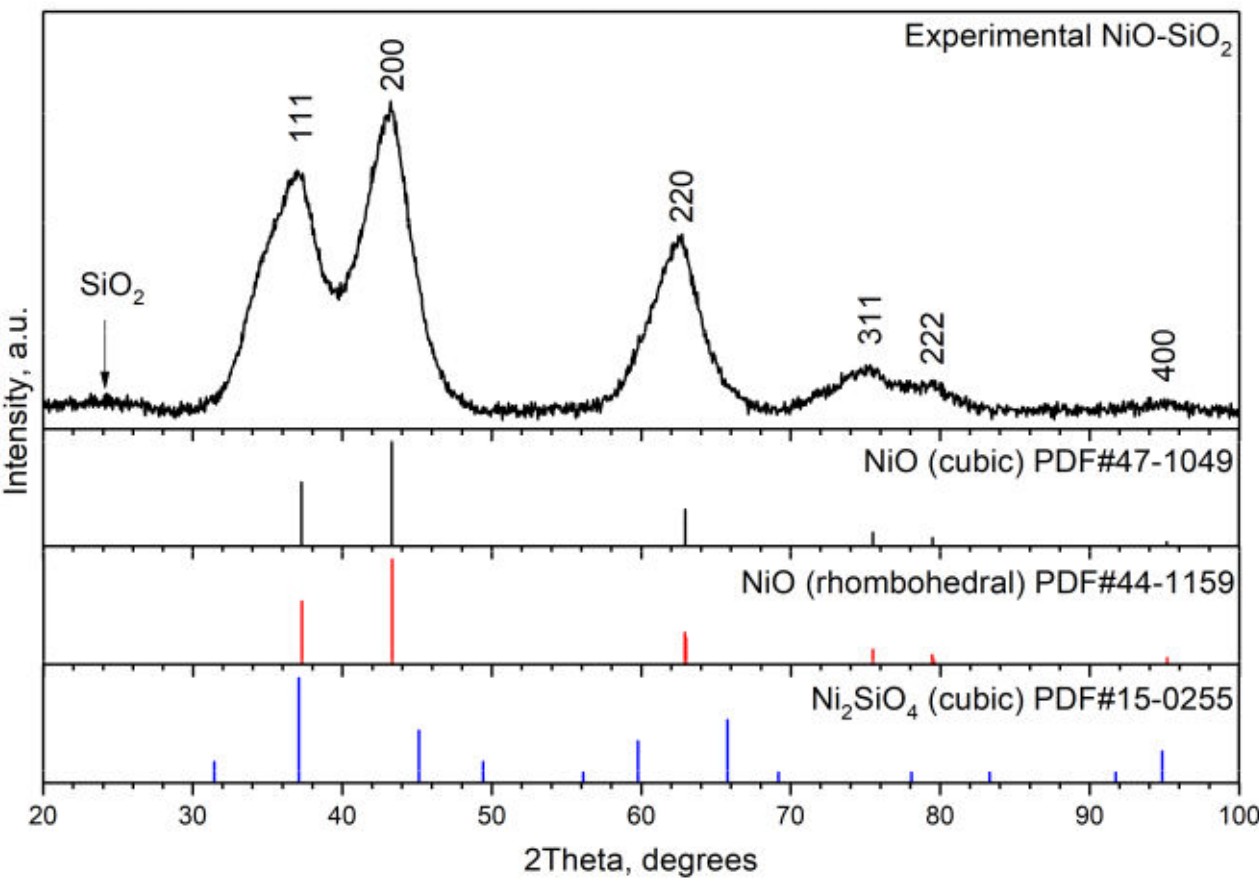

**Figure 1.** Experimental XRD pattern of NiO-SiO$_2$. Bars show the positions and intensities of the diffraction peaks of cubic (black) and rhombohedral (red) NiO as well as cubic Ni$_2$SiO$_4$ (blue). Miller indexes are given for cubic NiO.

**Table 1.** Results of the Rietveld refinement using different models. R$_{wp}$—weighted-profile R-factor.

| Model | Phase | Space Group | wt% | Lattice Constants, Å | D, Å | R$_{wp}$, % |
|-------|-------|-------------|-----|----------------------|------|-------------|
| a | NiO-cubic | Fm3m | 100 | a = 4.209(3) | 20 | 37 |
| b | NiO-rhombohedral | R-3m | 100 | a = 3.005(4), c = 7.193(2) | 20 | 24 |
| c | NiO-cubic | Fm3m | 61(1) | a = 4.169(5) | 25 | 36 |
|   | NiO-cubic | Fm3m | 39(1) | a = 4.288(7) | 25 |   |
| d | NiO-cubic | Fm3m | 66(1) | a = 4.218(2) | 20 | 21 |
|   | Ni$_2$SiO$_4$. | Fd3m | 34(1) | a = 8.28(2) | 15 |   |

Thus, none of the above models is suitable. In the small angle range (2θ < 30°), a halo is observed, probably due to amorphous SiO$_2$. So, it can be concluded that one part of silicon is contained in mixed Ni-Si oxide and another part is in SiO$_2$.

### 2.2. HRTEM Characterization

According to TEM data, large plate aggregates with sizes of ca. 100 nm are observed (Figure 3a). They consist of small primary lamellar particles with sizes of 20–30 Å in the [111] direction and 150–200 Å in the perpendicular direction (Figure 3b). Calculation of interplanar distances gives 2.5 Å and 2.1 Å, which correspond to interplanar distances d$_{111}$ and d$_{200}$ of NiO (PDF#47-1049, d$_{111}$ = 2.41 Å, d$_{200}$ = 2.09 Å). It can be seen from the HRTEM image that particles are moving in the [111] direction, and consequently, the atomic rows are "wavy." The HAADF STEM image and EDS mapping of some areas are displayed

in Figure 3c,d, respectively. The mapping shows the uniform distribution of Ni and Si cations over the NiO-SiO$_2$ catalyst. The Ni:Si:O ratio is 5:1:7 (Si constitutes about 17% of the cation content).

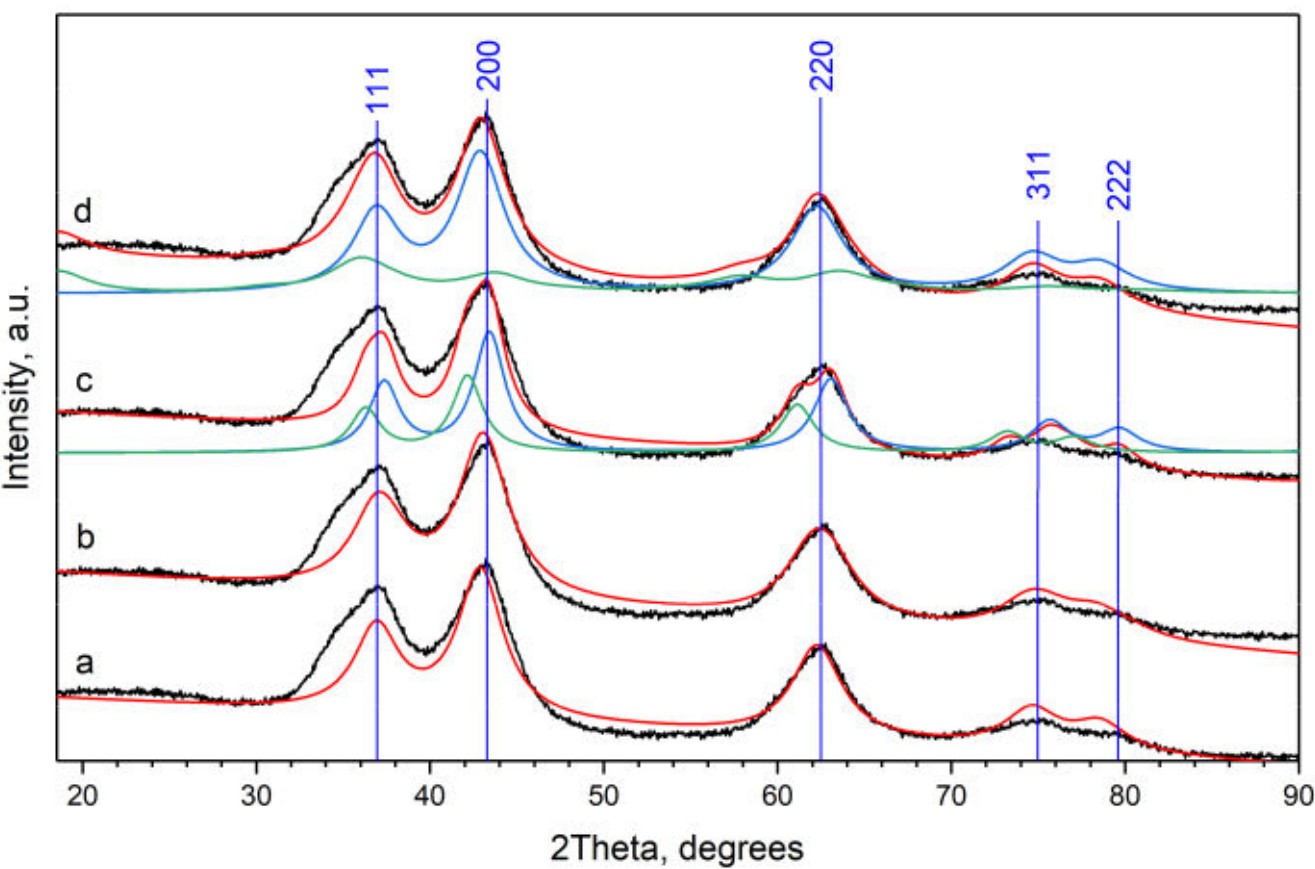

**Figure 2.** Rietveld analysis on the basis of (a) cubic NiO, (b) rhombohedral NiO, (c) two cubic NiO-like phases (blue and green), and (d) cubic NiO (blue) and cubic Ni2SiO4 (green). Experimental curves are black; calculated curves (the sum of XRD patterns calculated for individual phases and background) are red. Vertical blue lines—the positions of Bragg diffraction peaks for NiO.

### 2.3. Differential Dissolution

In contrast to XRD and TEM methods, which reveal structural and microstructural features, the differential dissolution method is able to determine the cationic composition of phases. This method determines the stoichiometry of the elemental composition of successively dissolving phases. The propagation of the dissolution reaction front from the surface to the center of the particles is accompanied by a continuous recording of the ratio between all elements.

Figure 4a shows the dissolution curves for Si and Ni depending on time, as well as the Si:Ni ratio. The dissolution of Si and Ni atoms begins simultaneously and, after the dissolution process, ends at the same time (Figure 4a). The dissolution of Ni initially proceeds at a low rate, and only with the increase in temperature (Figure 4a) does the dissolution rate increase until it reaches its maximum value in 25 min. From 25 to 40 min, the concentration of Ni drops, indicating the completion of its dissolution. Figure 4b shows the dissolution curves for Si and Ni, the Si:Ni ratio depending the on degree of Ni dissolution in at. %. From 25 to 90%, the profile of stoichiograms contains linear segments with a constant molar ratio of Si:Ni = 0.11. It indicates the presence of a solid solution of Si$_{0.11}$Ni$_1$ (since the oxygen content cannot be described by the DD method, the stoichiometric phase formulae are conventionally presented without oxygen). Note that during the first 15 min, the time-variable stoichiogram (the Si:Ni ratio, Figure 4a) shows an

increased value of Si:Ni = 0.2–0.5. The higher Si:Ni ratio in the initial dissolution period is due to the enrichment of the near-surface layers of this phase with silicon cations or to the formation of highly dispersed and strongly bonded $SiO_2$ particles. After subtracting the $Si_{0.11}Ni_1$ phase from the Si and Ni dissolution kinetic curves of the elements, the second (Si) and third (Ni) phases are distinguished. From these calculations, the total phase content is 85.7 wt% of $Si_{0.11}Ni_1$, 6.7 wt.% of Si, and 7.6 wt.% of Ni.

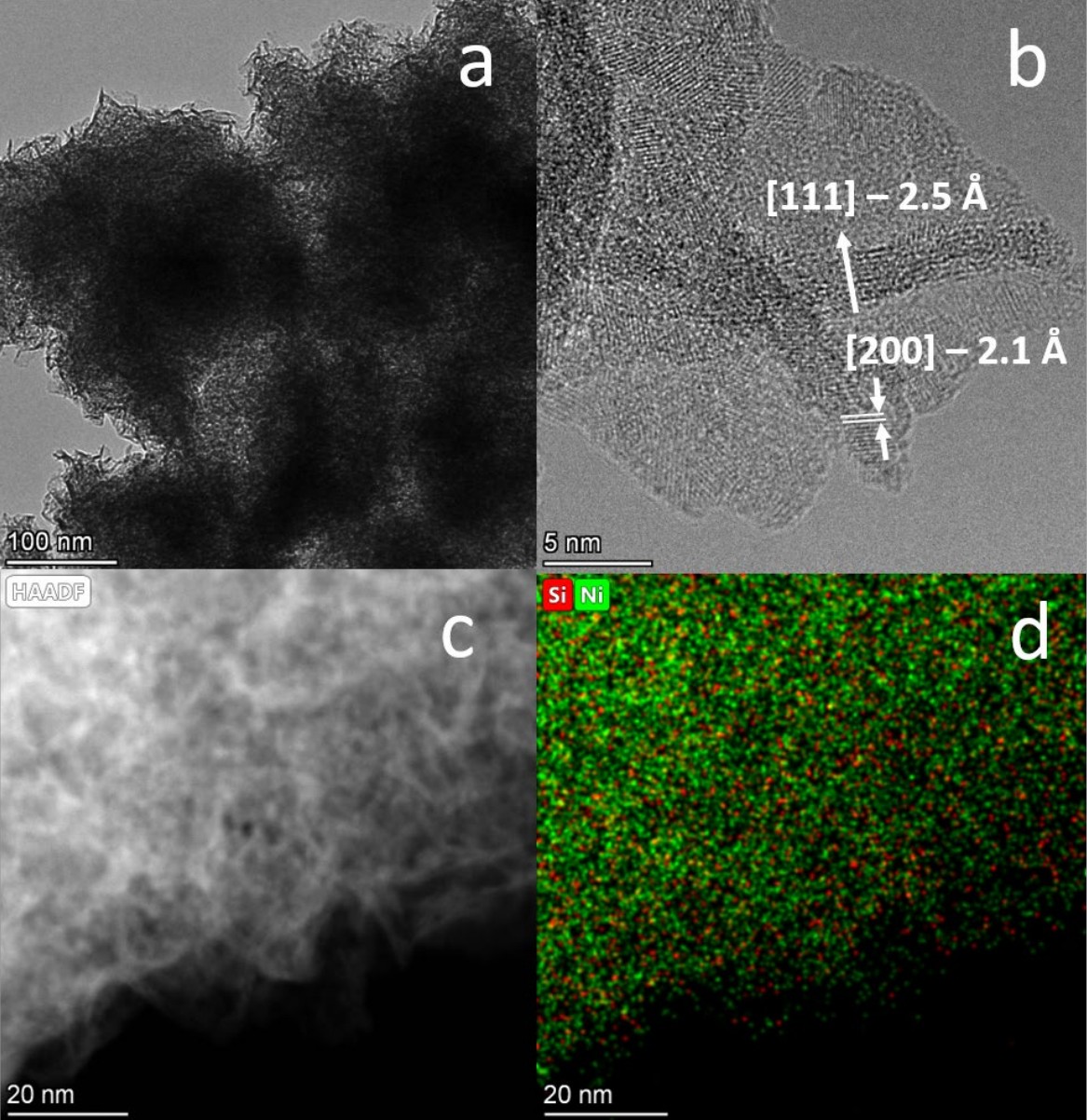

**Figure 3.** (**a**) TEM data; (**b**) HRTEM data; (**c**) the HAADF STEM image; (**d**) EDS mapping; Ni and Si atoms are green and red, respectively.

### 2.4. Simulation of SiO$_2$ Incorporation into the NiO Structure

Thus, the TEM and DD results indicate the presence of Si atoms in the structure of NiO. To create a model in which $Si^{4+}$ cations are included in the NiO structure, we proceeded from the cubic spinel structure of $Ni_2SiO_4$. The structures of NiO and $Ni_2SiO_4$ have different cationic sublattices and the same anionic sublattice, in which oxygen ions form a close packing (Figure 5). In spinel, $Ni^{2+}$ cations fill half of the octahedral voids, and 1/8 of the tetrahedral voids are occupied by $Si^{4+}$ cations. In the NiO structure, all octahedral voids are occupied by $Ni^{2+}$ cations, while there are no cations in tetrahedral voids.

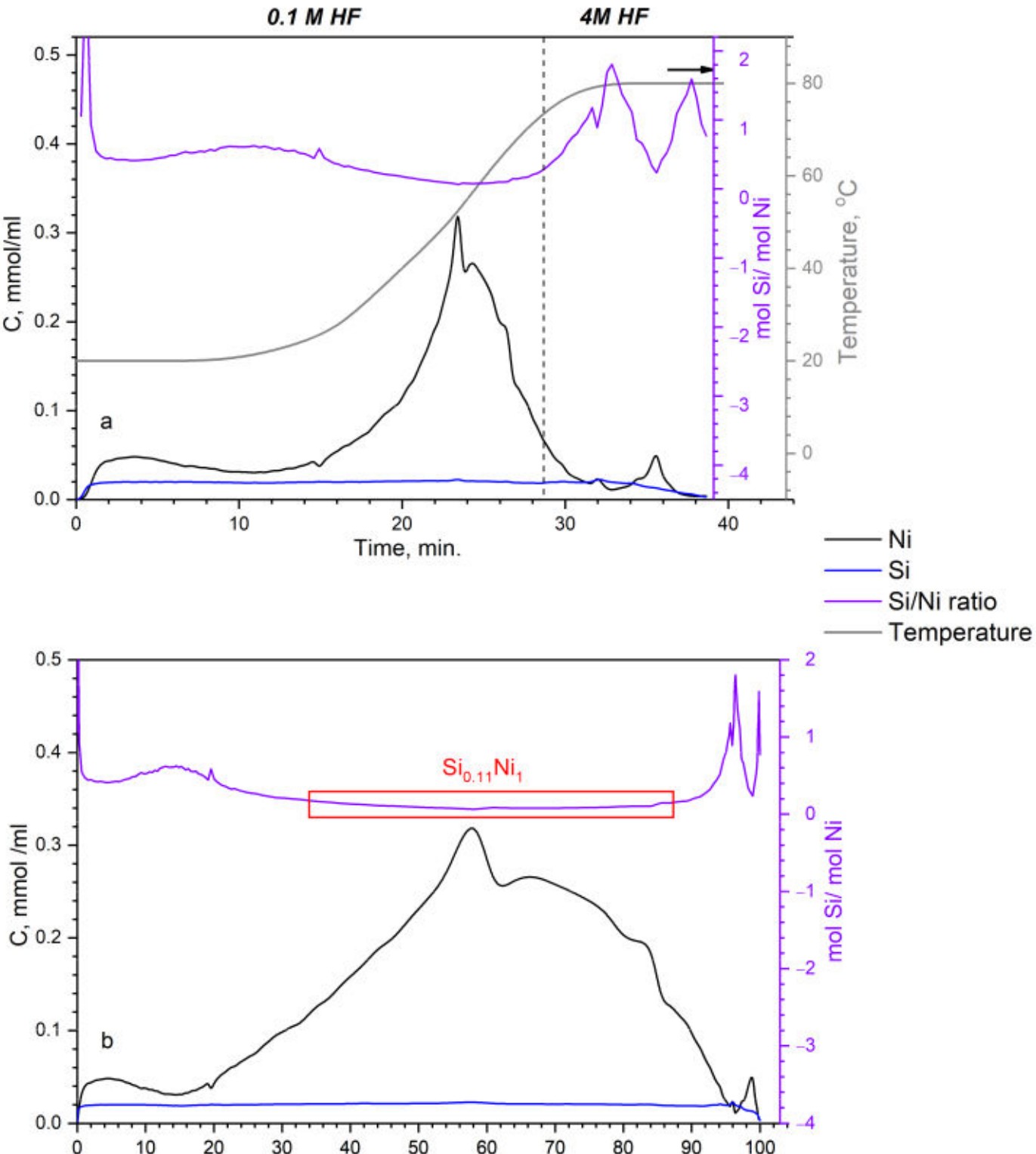

**Figure 4.** The dissolution curves for nickel and silicon and the Si/Ni ratio depend on time (**a**) and degree of Ni dissolution in at.% (**b**). The conditions of the dissolution depend on time (**a**). The red rectangle shows the area of constant Si:Ni ratio, equal to 0.11:1.

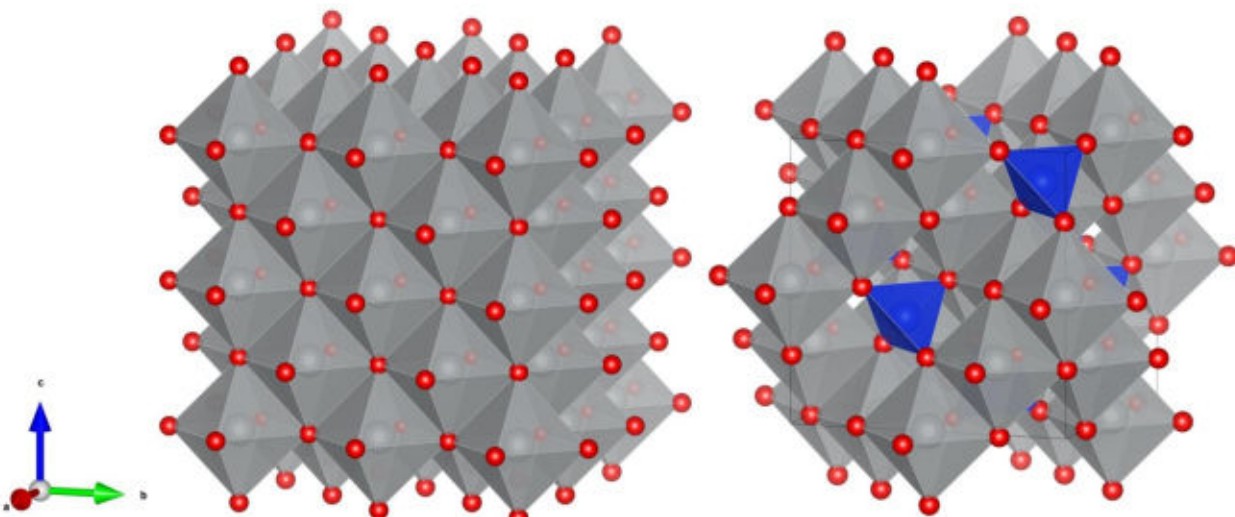

**Figure 5.** NiO structure (**left**); $Ni_2SiO_4$ structure (**right**) in cubic axes. Red ball indicates oxygen atom; grey—Ni, blue—Si.

If we consider the spinel structure along the [111] direction, which is normal for close-packed oxygen anions, then two types of layers can be distinguished: octahedral and mixed octahedral-tetrahedral. In the first one, $Ni^{2+}$ fills three of the four octahedral voids. In the mixed layer, two $Si^{4+}$ cations are in tetrahedral coordination, and a $Ni^{2+}$ cation is in the octahedral void. We modified these layers by removing $Ni^{2+}$ from the octahedral void in the mixed layer and by adding $Ni^{2+}$ to the empty octahedral void in the octahedral layer. In such a way, there are two types of layers: octahedral $Ni_4O_4$ and tetrahedral $Si_2O_4$ (Figure 6). The coordinates of the atoms were recalculated from a cubic unit cell to a hexagonal one with the use of equations $a_{hex} = a_{cubic}\sqrt{2}$; $c_{hex} = a_{cubic}/\sqrt{3}$. Therefore, the lattice constants are a = 5.9568 Å and c = 2.4319 Å. The coordinates of atoms are listed in Table 2. Additionally, each layer should be shifted by a vector (2/3, 1/3) in the plane of layers. This shift maintains the oxygen atoms' close packing. The particle shape was considered to be a cylinder with a variable height H and diameter D. The lamellar shape of the particles can be defined by D > H (the disk-like shape).

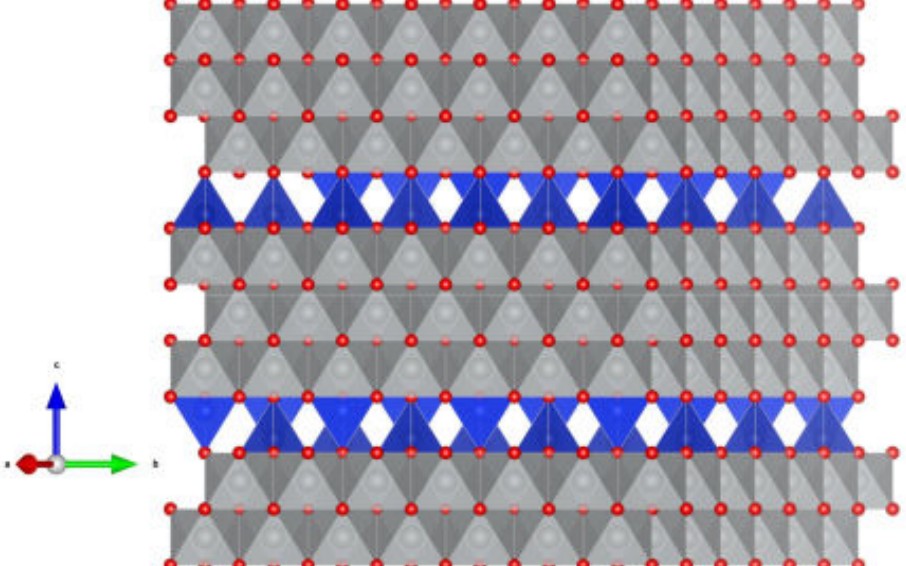

**Figure 6.** Model of the defect structure: incorporation of tetrahedral $Si_2O_4$ layers (blue) between octahedral $Ni_4O_4$ layers (grey) in hexagonal axes.

**Table 2.** Coordinates of atoms in hexagonal unit cells of octahedral $Ni_4O_4$ and tetrahedral $Si_2O_4$ layers.

| | $Ni_4O_4$ | | | | $Si_2O_4$ | | |
|---|---|---|---|---|---|---|---|
| **Atom** | **x** | **y** | **z** | **Atom** | **x** | **y** | **z** |
| O | 1/6 | 5/6 | 0 | O | 1/6 | 5/6 | 0 |
| | 1/6 | 1/3 | 0 | | 1/6 | 1/3 | 0 |
| | 2/3 | 5/6 | 0 | | 2/3 | 5/6 | 0 |
| | 2/3 | 1/3 | 0 | | 2/3 | 1/3 | 0 |
| Ni | 0 | 0 | 0.5 | Si | 1/3 | 2/3 | 0.25 |
| | 0.5 | 0 | 0.5 | | 2/3 | 1/3 | 0.75 |
| | 0 | 0.5 | 0.5 | | | | |
| | 0.5 | 0.5 | 0.5 | | | | |

We introduced statistically significant $Si_2O_4$ layers between $Ni_4O_4$ ones and calculated the corresponding XRD patterns (Figure 7). Different fractions ($\alpha$ = 0.0, 0.1, 0.2, 0.3 and 0.4) of $Si_2O_4$ layers were considered. So, the ratio $Si_2O_4$:$Ni_4O_4$ was changed from 0 to 0.4:0.6. Two crystal models of cylindrical particles consisted of 10 layers (height of the particle ~25 Å) with the diameter of 25 Å and 20 layers (height of the particle ~50 Å) with the diameter of 50 Å. The introduction of $Si_2O_4$ layers led to a decrease in intensity of all the peaks due to the smaller scattering ability of $Si_2O_4$ layers relative to $Ni_4O_4$ ones. A decrease in intensity of the 200, 220, 311 and 222 peaks took place without changes in their width and shape. As for the 111 peak, one can see that for smaller particle sizes (~25 Å) it became wider and shifted towards smaller diffraction angles. This could be explained by the appearance of additional diffuse scattering on the left side (smaller angles) of the 111 peak which is clearly observed for larger particle sizes(~50 Å).

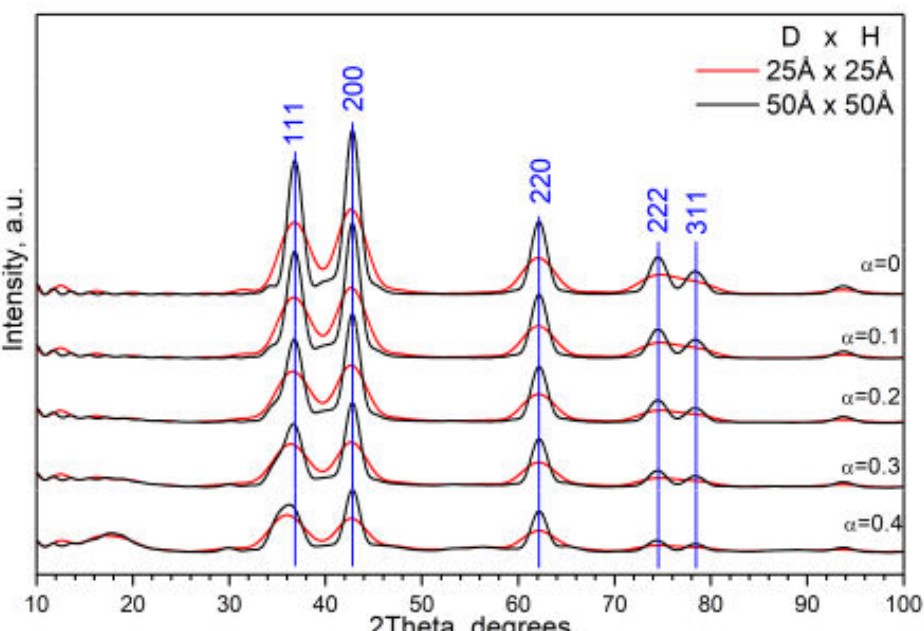

**Figure 7.** XRD patterns calculated for the statistical model consisting of $Ni_4O_4$ and $Si_2O_4$ layers with different fractions ($\alpha$) of $Si_2O_4$ layers. Calculations were carried out for cylindrical particles consisting of N = 10 layers (height of the particle H ~ 25 Å) with the diameter D ~ 25 Å and N = 20 layers (H ~ 50 Å) with D ~ 50 Å. Blue lines show Bragg positions for the 111, 200, 220, 331, and 222 peaks for NiO.

Thus, when layers of $Si_2O_4$ tetrahedra are added to the NiO structure, diffuse scattering appears on the left side of the 111 peak ($2\theta$ ~ 35°), which increases with an increase in the fraction of these layers. Depending on the particle size, this manifests itself either in the

asymmetry of the 111 peak (D ~ 50 Å) or in the effective shift of the 111-peak towards smaller angles (D ~ 25 Å).

Such changes in diffraction patterns when the $Si_2O_4$ tetrahedral layers are introduced between the $Ni_4O_4$ layers are in good correspondence with the experimental data. It can be assumed that the weak asymmetry of all the peaks is due to the presence of two NiO-like phases with different parameters in a cubic unit cell. Very high asymmetry of the 111 peak can be explained by the described above shift of the 111 peak toward smaller angles for very small crystallites (D ~ H ~ 25Å) consisting of octahedral $Ni_4O_4$ (NiO) and tetrahedral $Si_2O_4$ layers.

### 2.5. Rietveld Analysis by Two Cubic Phases

Since according to DD results, there are two Ni-containing phases (85.7 wt% of $Si_{0.11}Ni_1$ and 7.6 wt.% of Ni), we performed the Rietveld analysis based on two cubic NiO structures once again (Figure 8) in the 2θ angular range of 40–100° (for all the peaks except the first one). The lattice parameters were refined, and the average crystallite sizes were determined. In the main phase (~ 90 wt%), which has smaller average crystallite sizes (D ~ 20 Å), the lattice parameter a = 4.223Å was larger than that of bare NiO (PDF#47-1049, a = 4.177 Å). In the second phase with larger average crystallite sizes (D ~ 50 Å), the lattice parameter a = 4.188 Å was slightly larger compared to NiO. As expected, the first peak is poorly described by such a model. This is due to the fact that the introduction of $Si_2O_4$ tetrahedral layers between $Ni_4O_4$ layers was not specified here. This is to be done next.

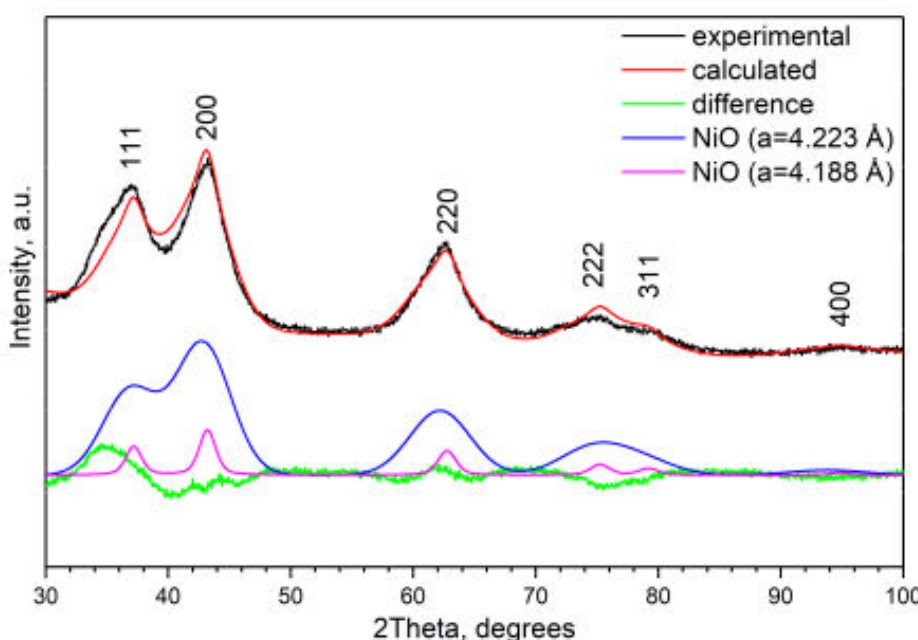

**Figure 8.** Rietveld analysis of two NiO cubic phases with different lattice constants (the first peak was excluded from the refinement).

### 2.6. Optimization of the Model of Cylindrical Particles Consisting of $Ni_4O_4$ and $Si_2O_4$ Layers

In order to simulate the structure of Ni-Si oxide particles, we subtracted the XRD pattern of NiO (a = 4.188 Å, D = 50 Å) from the experimental diffraction pattern (Figure 9). The remaining XRD pattern (Figure 9, the bottom black curve) corresponds to the Ni-Si oxide with an increased parameter (a = 4.223Å). We calculated XRD patterns based on different models of cylindrical particles (Figure 9). The almost isotropic cylindrical shape (height H = 27.5Å, diameter D = 27.5Å, grey color, Figure 9) of the particle consisting of only the octahedral $Ni_4O_4$ layers (without tetrahedral $Si_2O_4$ ones) was the first model. One can see that the first peak in the experiment (the black curve) is displaced relative to the calculated one (the gray curve). The positions of the first peak on the experimental and

calculated XRD patterns almost coincide when the probability of tetrahedral layers is equal to $\alpha$ = 0.25 (the bottom green curve).

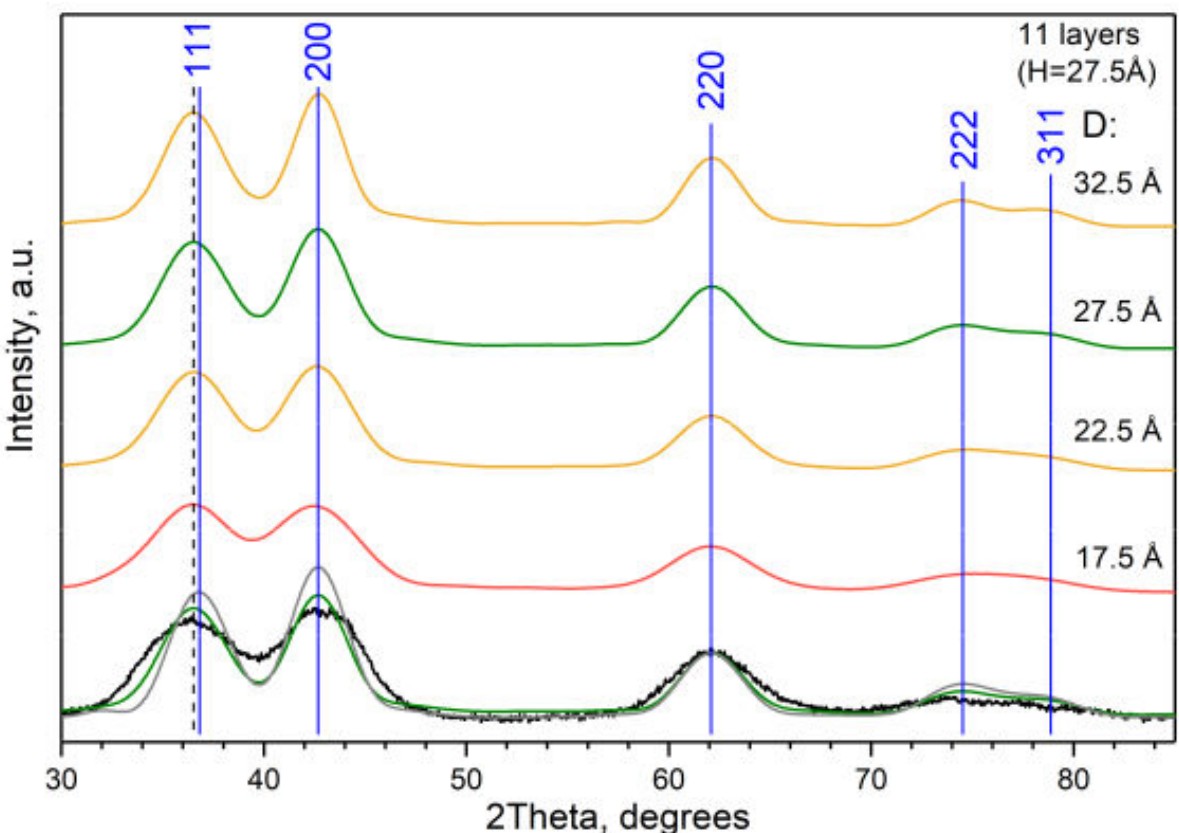

**Figure 9.** The partial experimental XRD pattern of mixed oxide (the black curve) obtained by subtraction of the XRD pattern calculated for NiO from the common experimental XRD pattern; XRD patterns calculated on the basis of different models of cylindrical particles (H the height, D the diameter). The fraction of $Si_2O_4$ layers was fixed at $\alpha$ = 0.25 (color curves). Grey curve was calculated for $\alpha$ = 0.0 (NiO with a = 4.223 Å), H = D = 27.5 Å. Blue lines show the positions of peaks for NiO ($\alpha$ = 0.0) with a = 4.223 Å. Dash line shows the position of the 111 peak at $\alpha$ = 0.25.

In the next step parameter $\alpha$ = 0.25 was fixed. The height (H) and diameter (D) of cylindrical particles were varied. Figure 9 shows the XRD patterns calculated for H = 27.5 Å and different diameters D. One can see that the XRD pattern calculated for sizes H = D = 27.5 Å and $\alpha$ = 0.25 (the green curve) has better correspondence with the experimental data relative to the XRD patterns calculated for the same sizes and $\alpha$ = 0.0 (the grey curve). Additionally, at $\alpha$ = 0.25, these sizes (H = D = 27.5 Å) give better correspondence relative to other ones (H = 27.5 Å, D = 17.5, 22.5 and 32.5 Å). In order to optimize the model parameters H and D, the following objective function was used

$$F = abs(I_{200}/I_{111} - 1.12) \qquad (1)$$

where $I_{111}$ and $I_{200}$ are the heights of the 111 and 200 peaks on the calculated XRD patterns, respectively. The experimental ratio $I_{200}/I_{111}$ is equal to 1.12. The objective function values for different particle sizes are presented in Table 3. One can see that there is a region with minimum values (green cells in Table 3). The best models have the following parameters: $\alpha$ = 0.25 for all the models: (1) H = D = 27.5 Å, (2) H = 25 Å, D = 30 Å and (3) H = 22.5 Å, D = 35 Å. Evidently, there is some distribution over both sizes of the particles. According to the TEM data (Figure 3), the particles are plate-like with sizes of 20–30 Å (the [111] direction) and 150–200 Å (the perpendicular direction). So, the average sizes in the [111] directions determined by XRD (height H) and TEM (thickness) coincide. As for the average sizes

in perpendicular directions, in all appearances, the whole particle cannot be considered a region of coherent scattering due to the "wavy" shape of the particles and should be divided into several such regions. In this context, there is no contradiction between TEM and XRD simulation results.

**Table 3.** Values of the objective function for different heights (H) and diameters (D) of cylindrical particles at the fixed fraction of $Si_2O_4$ layers ($\alpha = 0.25$).

| H, Å | D, Å | | | | | | | |
|------|------|------|------|------|------|------|------|------|
|      | 17.5 | 20 | 22.5 | 25 | 27.5 | 30 | 32.5 | 35 |
| 27.5 | 0.141 | 0.110 | 0.067 | 0.033 | 0.007 | 0.017 | 0.035 | 0.051 |
| 25 | 0.135 | 0.103 | 0.067 | 0.037 | 0.012 | 0.006 | 0.020 | 0.033 |
| 22.5 | x | 0.114 | 0.083 | 0.057 | 0.037 | 0.024 | 0.013 | 0.004 |
| 20 | x | x | 0.111 | 0.090 | 0.076 | 0.065 | 0.058 | 0.105 |
| 17.5 | x | x | x | 0.098 | 0.117 | 0.111 | 0.108 | 0.105 |
| 15 | x | x | x | x | 0.179 | 0.177 | 0.178 | 0.177 |

According to the 1D-disordered model XRD simulation, the concentration of $Si_2O_4$ layers in the NiO structure is equal to $\alpha = 0.25$. It indicates that the Si content is 14 at.% ($Si_{0.14}Ni_{0.86}O$), which is quite close to the value obtained from the DD method, 10 at.% ($Si_{0.1}Ni_{0.9}O$). The TEM data also show the even spread of Si over the sample. Simulation of XRD data shows that the sample contains ~90 wt% of mixed Ni-Si oxide ($Si_{0.14}Ni_{0.86}O$) with the average crystallite sizes (D ~ 22.5 27.5 Å) and cubic parameters equal to a = 4.223 Å and ~10% of pure NiO (D ~ 50 Å) with a = 4.188 Å. These results are also supported by the DD data, which showed that the sample consists of three phases: 7.6 wt.% of Ni, 85.7 wt.% of $Si_{0.1}Ni_{0.9}$, and 6.7 wt% of Si. The amorphous Si oxide is also observed on the XRD pattern as the halo at 2θ angles lower than 30° (Figure 1).

## 3. Materials and Methods

Solid nickel (II) carbonate basic hydrate was mixed with the required amounts of an aqueous solution of ammonia (25% of $NH_3$) and distilled water. Then, ethyl silicate (with a $SiO_2$ content of 32 wt%) was added to the solution. The obtained suspension was dried in air for 12 h at 115 °C and then calcined at 400 °C for 4 h. The X-ray fluorescence analysis shows the atomic ratio in the synthesized sample is equal to Si:Ni = 1:4.

XRD data of the sample were obtained using a D8 Advance X-ray diffractometer (Bruker, Berlin, Germany), with monochromatic Cu-Kα radiation (λ = 1.5418 Å). The diffractometer was equipped with a LynxEye 1D detector (Bruker, Berlin, Germany), which allowed us to obtain a diffraction pattern in the 2θ range from 10 to 100°, with a step of 0.05° and accumulation time 15 s per point.

Rietveld refinement for quantitative analysis was carried out using the software package TOPAS V.4.2. The average crystallite size was calculated as LVol-IB values (i.e., the volume weighted mean column height based on integral breadth).

Simulation of XRD patterns for the mixed Ni-Si oxide was carried out on the basis of models of one-dimensionally (1D) disordered crystals with the use of software [27]. The model was built as a statistical sequence of a finite number (N) of two kinds of 2D periodic layers ($Ni_4O_4$ and $Si_2O_4$). First, the scattering amplitudes were calculated along infinitely narrow hk rods for each kind of layer of infinite size. Then, the scattering intensity distribution along hk rods for the statistical sequence of layers was calculated. A Markov chain was used as a statistical rule for the generation of the sequence. A variable parameter was the fraction ($\alpha$) of $Si_2O_4$ layers. Consequently, the fraction of $Ni_4O_4$ layers was equal to $1 - \alpha$. The diffraction effect of the layer shape (a circle of diameter D) was taken into account by the convolution of rod intensity with the squared module of the Fourier transform of the shape function. The total distribution of the scattering intensity was calculated for the ensemble of chaotically oriented cylindrical stacks of layers (cylindrical crystallites). The

simulated XRD patterns were compared with the experimental ones visually and using the objective function (1).

HRTEM images were obtained using a ThemisZ Thermo Fisher Scientific microscope (Thermo Fisher Scientific, Eindhoven, Netherlands) with a resolution of 0.7 Å, respectively. Elemental maps were obtained using energy dispersive spectrometer SuperX Thermo Fisher Scientific. Samples for research were fixed on standard copper grids using ultrasonic dispersion of the catalysts in ethanol.

To determine the qualitative and quantitative phase compositions, the DD method was used. The sample was analyzed in a flow reactor in the stoichiographic titration mode, successively changing the solvent concentration from 0.1 to 4 M HF and the temperature from 20 to 80 °C. It should be noted that the sample is dissolved completely under these conditions. The composition of the solution at the reactor outlet was determined by atomic emission spectroscopy on a PST device (BAIRD, The Netherlands) using the following spectral lines of the elements: Ni—231.6 nm and Si—288.1 nm; the accuracy of the concentration measurements was 5%; the sensitivity level—$10^{-9}$ g/mL.

## 4. Conclusions

The structure of the Ni-Si oxide prepared by the sol-gel method followed by calcination at 400 °C was determined first. Rietveld refinement was not applicable for the structure analysis of the mixed oxide due to its imperfect structure. Simulation of the XRD patterns on the basis of statistical models of 1D disordered crystals allowed us to discover that the Ni-Si oxide consists of octahedral $Ni_4O_4$ (or NiO) and tetrahedral $Si_2O_4$ (or $SiO_2$) layers. It was shown that the introduction of $Si_2O_4$ layers between $Ni_4O_4$ layers leads to the shift of the 111 peak towards smaller angles, while the positions of other peaks remain unchanged. In addition, redistribution of the peaks' relative intensities takes place. At a definite ratio of $Ni_4O_4:S_2O_4 = 0.75:0.25$ (or Ni:Si = 0.86:0.14), good correspondence with the experimental XRD pattern was achieved. According to the DD, the Ni:Si oxide has a ratio of Ni:Si = 1:0.11 (or Ni:Si = 0.90:0.10). HRTEM combined with the EDX showed that the Si is homogeneously distributed over platelet NiO-like particles. The layered structure of mixed Ni-Si oxide and platelet particle shape can be inherited from the layered double Ni-Si hydroxide precursor.

**Author Contributions:** Conceptualization, V.A.Y. and O.A.B.; methodology; S.V.C., A.A.P. and E.Y.G.; software, S.V.C.; formal analysis, M.D.M., M.V.A. and R.G.K.; investigation, M.V.A., R.G.K. and M.D.M.; writing—original draft preparation, M.D.M., S.V.C. and O.A.B.; writing—review and editing, M.D.M., O.A.B., S.V.C. and E.Y.G.; visualization M.D.M. and S.V.C.; supervision V.A.Y. All authors have read and agreed to the published version of the manuscript.

**Funding:** This research was funded by the Ministry of Science and Higher Education of the Russian Federation, projects AAAA-A21-121011390011-4, AAAA-A21-121011390007-7, and AAAA-A21-121011390053-4.

**Informed Consent Statement:** Not applicable.

**Acknowledgments:** The studies were carried out using the facilities of the shared research center "National center of investigation of catalysts" at the Boreskov Institute of Catalysis.

**Conflicts of Interest:** The authors declare no conflict of interest.

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
