# Peer review of "Defect Structure of Nanocrystalline NiO Oxide Stabilized by SiO2"

_inorganics, doi:10.3390/inorganics11030097_

Round 1

Reviewer 1 Report

1. The language needs to be improved. There are typos such as "with sizes of 20÷30 Å".

2. For figure 1, the bar hight for NiO and Ni2SiO4 should be corrected. The peak for SiO2 should be given for reference.

3. The author claimed that the atomic ratio in the synthesized sample equal to Si:Ni=1:4, but also claimed a Ni-Si oxide catalyst (Ni : Si = 84 : 16 mol.%) was obtained by sol-gel 323 method followed by calcination at 400°C for 1 hour. Please check.

4. Please check this  statement "HR TEM combined with EDX showed that Si is homogeneously distributed over NiO-like particles having platelet shape with size of about 20-30 Å in 330 [111] direction and 150-200 Å in normal to [111] directions".

5. It is known that SiF4 is in gas phase at room temperature.  How did the author make conclusion about Si/Ni from DD?

Author Response

  1. The language needs to be improved. There are typos such as "with sizes of 20÷30 Å".

According to reviewer recommendation, the English was improved.

  1. For figure 1, the bar hight for NiO and Ni2SiO4 should be corrected. The peak for SiO2 should be given for reference.

According to reviewer recommendation, Figure 1 was corrected. The bars of the positions and intensities of the diffraction peaks of cubic (black) and rhombohedral (red) NiO as well as cubic Ni2SiO4 (blue) was moved down.

  1. The author claimed that the atomic ratio in the synthesized sample equal to Si:Ni=1:4, but also claimed a Ni-Si oxide catalyst (Ni : Si = 84 : 16 mol.%) was obtained by sol-gel 323 method followed by calcination at 400°C for 1 hour. Please check.

It was mistake. We have changed the conclusions and deleted this sentence.

  1. Please check this statement "HR TEM combined with EDX showed that Si is homogeneously distributed over NiO-like particles having platelet shape with size of about 20-30 Å in 330 [111] direction and 150-200 Å in normal to [111] directions".

To improve clarity of manuscript we have changed the conclusions and deleted this sentence.

  1. It is known that SiF4 is in gas phase at room temperature. How did the author make conclusion about Si/Ni from DD?

In the Differential Dissolution method, the dissolution process of a sample occurs in a closed flow reactor. In the flow reactor, solid substance interacts with fresh portion of solvent which does not contain dissolution products. The temperature and concentration of solvent are increasing over time. Large excess of solvent is needed to increase the concentration despite part of it is used for the interaction with the sample.

The HF has ability to interact with silicon oxide (IV) with the formation of H2SiF6. The significant amount of silicon is stored in the solution in this state. Probably, this is also facilitated by the property of water to absorb SiF4, as well as the ability of SiF4 to attach additional fluorine ions with the formation of complex fluorosilicate ions that are stable in an aqueous solution, for example:

SiF4 + 2F- → SiF62-

or the equation in the general:

SiO2 + 6HF ↔ H2[SiF6] + 2H2O

Additionally, the SiF4 loss was prevented by adding into the water solution of HF boric acid solution with concentration 1.5%.

The DD results are compared with independent data of the elemental analysis of the sample; the X-Ray fluorescence analysis was used in our work.

Reviewer 2 Report

In this paper, the structural features of NiO-SiO2 nanocrystalline catalyst were studied by XRD and TEM, and the defect structure and composition of Ni-Si oxides were revealed. The results have some reference value for the analysis of SiO2 loaded Ni-based materials. The framework and content of the article are reasonable, and the structure analysis is in place, so it is recommended to publish. But there is a small problem can be improved.

1. In Fig.4, there are multiple curves and symbols in the figure (for example, two short thick vertical lines in b). Legend should be added to the figure to explain what each line and symbols represents.

Author Response

According to reviewer recommendation, Figure 4 was corrected. Abundant symbols were deleted and Legend was added.

Reviewer 3 Report

The authors presented an interesting study concerning the recognition and interpretation of the structural features of NiO-SiO2 nanocrystalline system. The simulation results provide additional insight into the host-guest composition, the internal microstructure of the Si-doped NiO-based materials, and the crystal chemical features of the ordering of SiO2 tetrahedral layers between NiO octahedral layers in the complex lattice structure of mixed Ni-Si oxide. The paper is complete, well-organized and properly referenced. The simulations are clearly described. Experimental data are presented and discussed in an understandable manner. The analysis and conclusions are consistent and well supported by data and explanations. Thus, this manuscript is worthy of publication.

Author Response

We are thankful to the referee for attentive reading and high evaluation. 

Reviewer 4 Report

The manuscript presented by Maxim D. Mikhnenko and co-authors about Defect Structure of Nanocrystalline NiO Oxide Stabilized by SiO2, the manuscript looks interesting. The authors should know when to make the definition for new phrase, for example, the definition of differential dissolution (DD) should be done for the first time in the abstract and content, since no one knows what is DD (DD method, DD results, DD data) without the definition. The authors should show clearly the novelty for this study, especially in the abstract and conclusions, but the abstract and conclusion are just like data analysis and simple description, more scientific discussion should be added. This manuscript focus on the XRD analysis, and thus the authors should show their characteristics, to show what was the difference from others and to attract the readers. The format should be unified for the tables and figures. The images with higher resolution should be made if it is possible. Some sentences were written not so well, the English should be improved, and there are some small mistakes could be found in the content and references, the mistakes should be corrected. All the references should be unified and followed the requirements of this journal (in the reference part, some reference with DOI but others without DOI, the subscript should be used for the formula in the reference, for example, CO2, O2, CeO2, SiO2 and so on), the authors should check these carefully.

Author Response

  1. The authors should know when to make the definition for new phrase, for example, the definition of differential dissolution (DD) should be done for the first time in the abstract and content, since no one knows what is DD (DD method, DD results, DD data) without the definition

According to reviewer recommendation, the definition of differential dissolution (DD) was done for the first time in the abstract

  1. The authors should show clearly the novelty for this study, especially in the abstract and conclusions, but the abstract and conclusion are just like data analysis and simple description, more scientific discussion should be added. This manuscript focus on the XRD analysis, and thus the authors should show their characteristics, to show what was the difference from others and to attract the readers

According to reviewer recommendation we have changed the abstract and conclusions.

  1. The format should be unified for the tables and figures. The images with higher resolution should be made if it is possible.

According to reviewer recommendation the style of tables was corrected. Figures 1 and 4 were changed.

  1. Some sentences were written not so well, the English should be improved, and there are some small mistakes could be found in the content and references, the mistakes should be corrected.

According to reviewer recommendation, the mistakes and typos in the English, references were corrected.

  1. All the references should be unified and followed the requirements of this journal (in the reference part, some reference with DOI but others without DOI, the subscript should be used for the formula in the reference, for example, CO2, O2, CeO2, SiO2 and so on), the authors should check these carefully.

According to viewer recommendation the mistakes and typos in the reference list were corrected.

Round 2

Reviewer 1 Report

This work has been improved and could be published now.

Reviewer 4 Report

The authors had made the revision based on the comments. The present manuscript looks much better now after the modification, and thus this it is recommended for publication. Additionally, I suggest the authors had better mark the revised part as different color, and thus it easy for reviewers to know which parts were modified, otherwise, we have to compare the present and previous version together to look for the difference. There are still some small mistakes can be found fond in the manuscript, for example, the format of reference 31 was different from others, the unit of “ml” should be “mL” in the content and so on. The authors should check the manuscript carefully before the final publication.